# Learning to Extend Molecular Scaffolds with Structural Motifs

**Krzysztof Maziarz**[*]
Microsoft Research
United Kingdom

**Henry Jackson-Flux**
Microsoft Research
United Kingdom

**Pashmina Cameron**
Microsoft Research
United Kingdom

**Finton Sirockin**
Novartis
Switzerland

**Nadine Schneider**
Novartis
Switzerland

**Nikolaus Stiefl**
Novartis
Switzerland

**Marwin Segler**
Microsoft Research
United Kingdom

**Marc Brockschmidt**
Microsoft Research
United Kingdom

## Abstract

Recent advancements in deep learning-based modeling of molecules promise to accelerate *in silico* drug discovery. A plethora of generative models is available, building molecules either atom-by-atom and bond-by-bond or fragment-by-fragment. However, many drug discovery projects require a fixed scaffold to be present in the generated molecule, and incorporating that constraint has only recently been explored. Here, we propose MoLeR, a graph-based model that naturally supports scaffolds as initial seed of the generative procedure, which is possible because it is not conditioned on the generation history. Our experiments show that MoLeR performs comparably to state-of-the-art methods on unconstrained molecular optimization tasks, and outperforms them on scaffold-based tasks, while being an order of magnitude faster to train and sample from than existing approaches. Furthermore, we show the influence of a number of seemingly minor design choices on the overall performance.

## 1 Introduction

The problem of *in silico drug discovery* requires navigating a vast chemical space in order to find molecules that satisfy complex constraints on their properties and structure. This poses challenges well beyond those solvable by brute-force search, leading to the development of more sophisticated approaches. Recently, deep learning models are becoming an increasingly popular choice, as they can discern the nuances of drug-likeness from raw data.

While early generative models of molecules relied on the textual SMILES representation and reused architectures from natural language processing (Segler et al., 2018; Gómez-Bombarelli et al., 2018; Winter et al., 2019a; Ahn et al., 2020), many recent approaches are built around molecular graphs (De Cao & Kipf, 2018; Liu et al., 2018; Li et al., 2018b; Assouel et al., 2018; Simonovsky & Komodakis, 2018; Jin et al., 2018; 2020; Bradshaw et al., 2020). Compared to SMILES-based methods, graph-based models that employ a sequential generator enjoy perfect validity of generated molecules, as they can enforce hard chemical constraints such as valence during generation.

However, even if a molecule does not violate valence constraints, it is merely a sign of *syntactic* validity; the molecule can still be *semantically* incorrect by containing unstable or unsynthesisable substructures. Intermediate states during atom-by-atom generation may contain atypical chemical fragments, such as alternating bond patterns corresponding to unfinished aromatic rings. Therefore, some works (Rarey & Dixon, 1998; Jin et al., 2018; 2020; Ståhl et al., 2019; Xie et al., 2021) propose data-driven methods to mine common molecular fragments – referred to as *motifs* – which can be used to build molecules fragment-by-fragment instead of atom-by-atom. When motifs are employed, most partial molecules during generation are semantically sensible, as they do not contain half-built structures such as partial rings.

---

[*]Correspondence to krzysztof.maziarz@microsoft.com

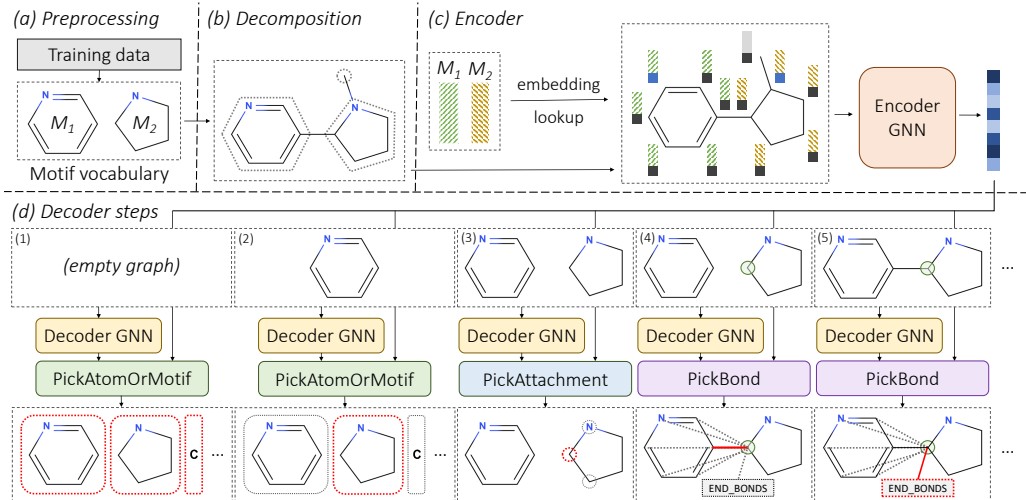

Figure 1: Overview of our approach. We discover motifs from data (a) and use them to decompose an input molecule (b) into motifs and single atoms. In the encoder (c), atom features (bottom) are combined with motif embeddings (top), making the motif information available at the atom level. Decoder steps (d) are only conditioned on the encoder output and partial graph (hence independent) and have to select one of the valid options (shown below, correct choices marked in red).

A common additional constraint in drug discovery projects is the inclusion of a predefined subgraph, called a *scaffold* (Schuffenhauer et al., 2007). Sampling molecules that contain a given scaffold can be approached by unconditional generation followed by post-hoc filtering. While simple, this method is not scalable, as the number of samples required may grow exponentially with scaffold size. Instead, some recent models can enforce the presence of a given scaffold (Lim et al., 2019; Li et al., 2019; Arús-Pous et al., 2020; Langevin et al., 2020). However, extending an arbitrary generative model to perform scaffold-based generation is often non-trivial, as we discuss in Section 4.

In this work we make the following contributions:

- In Section 2 we present MoLeR, a new graph-based generative model suitable for the commonly required task of extending partial molecules. It can use motifs (molecule fragments) to generate outputs (similarly to Jin et al. (2018; 2020)), but integrates this with atom-by-atom generation.
- We show experimentally in Section 3 that MoLeR *(a)* is able learn to generate molecules matching the distribution of the training data (with and without scaffolds); *(b)* together with an off-the-shelf optimization method (MSO (Winter et al., 2019b)) can be used for molecular optimization tasks, matching the state of the art methods in unconstrained optimization, and outperforming them on scaffold-constrained tasks; and *(c)* is faster in training and inference than baseline methods.
- We also perform experiments in Section 3 to analyze two design decisions that are understudied in the literature: the choice of the generation order and the size of the motif vocabulary. Our results show how varying these two parameters affects model performance.

Code is available at `https://github.com/microsoft/molecule-generation`.

## 2   OUR APPROACH

### 2.1   DATA REPRESENTATION

**Motifs**    Training our model relies on a set of fragments $\mathcal{M}$ – called the *motif vocabulary* – which we infer directly from data. For each training molecule, we decompose it into fragments by breaking some of the bonds; as breaking rings is chemically challenging, we only consider *acyclic bonds*, i.e. bonds that do not lie on a cycle. We break all acyclic bonds adjacent to a cycle (i.e. at least one endpoint lies on a cycle), as that separates the molecule into cyclic substructures, such as ring systems, and acyclic substructures, such as functional groups. We then aggregate the resulting fragments over the entire training set, and define $\mathcal{M}$ as the $n$ most common motifs, where $n$ is a hyperparameter. Having selected $\mathcal{M}$, we pre-process molecules (both for training and during inference) by noting which atoms are covered by motifs belonging to the vocabulary. This is done by applying the same

bond-breaking procedure as used for motif vocabulary extraction. During generation, our model can either add an entire motif in one step, or generate atoms and bonds one-by-one. This means that it can generate arbitrary structures, such as an unusual ring, even if they do not appear in the training data.

Finally, note that in contrast to Jin et al. (2020), we do not decompose ring systems into individual rings. This means that our motifs are atom-disjoint, and we consequently do not need to model a motif-specific attachment point vocabulary, as attaching a motif to a partial graph requires adding only a single bond, and thus there is only one attachment point.

**Molecule Representation**    We represent a molecule as a graph $\mathcal{G} = (\mathcal{V}, \mathcal{E})$, where vertices $\mathcal{V}$ are atoms, and edges $\mathcal{E}$ are bonds. Edges may also be annotated with extra features such as the bond type. Each node (atom) $v \in \mathcal{V}$ is associated with an initial node feature vector $h_v^{(init)}$, chosen as chemically relevant features (Pocha et al., 2020), both describing the atom (type, charge, mass, valence, and isotope information) and its local neighborhood (aromaticity and presence of rings). These features can be readily extracted using the RDKit library (Landrum et al., 2006). Additionally, for atoms that are part of a motif, we concatenate $h_v^{(init)}$ with the motif embedding; for the other atoms we use a special embedding vector to signify the lack of a motif. We show this at the top of Figure 1.

Throughout this paper we use Graph Neural Networks (Li et al., 2015; Kipf & Welling, 2016) to learn contextualized node representations $h_v$ (see Appendix A for background information on GNNs). Motif embeddings are initialized randomly, and learned end-to-end with the rest of the model.

## 2.2    THE MoLeR DECODER

Our generative procedure is shown in Algorithm 1 and example steps are shown at the bottom of Figure 1.  It takes as input a conditioning input vector $z$, which can either be obtained from encoding (in our setting of training it as an autoencoder) or from sampling (at inference time), and optionally a partial molecule to start generation from.  Our generator constructs a molecule piece by piece. In each step, it first selects a new atom or entire motif to add to the current partial molecule, or to stop the generation. If generation continues, the new atom (or an atom picked from the added motif) is then in "focus" and connected to the partial molecule by adding one or several bonds.

Our decoder relies on three neural networks to implement the functions PickAtomOrMotif,

---

**Algorithm 1** MoLeR's Generative Procedure

**Input:** vector $z$, scaffold as partial graph $S$
**Output:** molecule $M$ and probability $p$
$M, p \leftarrow S, 1$
**while** True **do**
    $a, p^a \leftarrow$ PickAtomOrMotif$(z, M)$
    $p \leftarrow p \cdot p^a$
    **if** $a =$ END_GEN **then**
        **return** $M, p$
    $M \leftarrow$ AddAtomOrMotif$(M, a)$
    $v^\circledcirc, p^\circledcirc \leftarrow$ PickAttachment$(z, M, a)$
    $p \leftarrow p \cdot p^\circledcirc$
    **while** True **do**
        $b, p^b \leftarrow$ PickBond$(z, M, v^\circledcirc)$
        $p \leftarrow p \cdot p^b$
        **if** $b =$ END_BONDS **then**
            **break**
        $M \leftarrow$ AddBond$(M, b)$

---

PickAttachment and PickBond. These share a common GNN to process the partial molecule $M$, yielding high-level features $h_v$ for each atom $v$ and an aggregated graph-level feature vector $h_{mol}$. We call our model MoLeR, as each step is conditioned on the underlined Molecule-Level Representation $h_{mol}$.

PickAtomOrMotif uses $h_{mol}$ as an input to an MLP that selects from the set of known atom types, motifs, and a special END_GEN class to signal the end of the generation. PickAttachment is used to select which of the atoms in an added motif to connect to the partial molecule (this is trivial in the case of adding a single atom). This is implemented by another MLP that computes a score for each added atom $v_a$ using its representation $h_{v_a}$ and $h_{mol}$. As motifs are often highly symmetric, we determine the symmetries using RDKit and only consider one atom per equivalence class. An example of this is shown in step (3) at the bottom of Figure 1, where only three of the five atoms in the newly-added motif are available as choices, as there are only three equivalence classes.

Finally, PickBond is used to predict which bonds to add, using another MLP that scores each candidate bond between the focus atom $v^\circledcirc$ and a potential partner $v_b$ using their representations $h_{v^\circledcirc}$, $h_{v_b}$ and $h_{mol}$. We also consider a special, learned END_BONDS partner to allow the network to choose to stop adding bonds. Similarly to Liu et al. (2018), we employ valence checks to mask out bonds that would lead to chemically invalid molecules. Moreover, if $v^\circledcirc$ was selected as an attachment point in a motif, we mask out edges to other atoms in the same motif.

The probability of a generation sequence is the product of probabilities of its steps; we note that the probability of a molecule is the sum over the probability of all different generation sequences leading to it, which is infeasible to compute. However, note that steps are only conditioned on the input $z$ and the current partial molecule $M$. Our decoding is therefore not fully auto-regressive, as it marginalizes over all different generation sequences yielding the partial molecule $M$. During training, we use a softmax over the candidates considered by each subnetwork to obtain a probability distribution. As there are many steps where there are several correct next actions (e.g. many atoms could be added next), during training, we use a multi-hot objective that encourages the model to learn a uniform distribution over all correct choices. For more details about the architecture see Appendix B.

## 2.3 MOLECULE GENERATION ORDERS

As alluded to above, to train MoLeR we need to provide supervision for each individual step of the generative procedure, which is complicated by the fact that a single molecule may be generated in a variety of different orders. To define a concrete generation sequence, we first choose a starting atom, and then for every partial molecule choose the next atom from its *frontier*, i.e. atoms adjacent to already generated atoms. After each choice, if the currently selected atom is part of a motif, we add the entire motif into the partial graph at once. We formalize this concept in Algorithm 2.

In Section 3, we evaluate orders commonly used in the literature: *random*, where ValidFirstAtoms returns all atoms and ValidNextAtoms all atoms

---

**Algorithm 2** Determining a generation order

**Input:** Target molecule $M$, partial mapping $\mathcal{A}$ from atoms to motifs that cover them
$t, V_0 \leftarrow 0, \emptyset$
**while** not all atoms visited **do**
    **if** $t = 0$ **then**
        $c_t \leftarrow$ ValidFirstAtoms$(M)$
    **else**
        $c_t \leftarrow$ ValidNextAtoms$(V_t, M)$
    $a_t \sim \mathcal{U}(c_t)$       ▷ Sample $a_t$ uniformly
    **if** $a_t$ is covered by $\mathcal{A}$ **then**
        $V_+ \leftarrow \mathcal{A}(a_t)$   ▷ Add an entire motif
    **else**
        $V_+ \leftarrow \{a_t\}$     ▷ Add a single atom
    $V_{t+1}, t \leftarrow V_t \cup V_+, t + 1$

---

on the frontier on the current partial graph, i.e., a randomly chosen valid generation order; *canonical*, which is fully deterministic and follows a canonical ordering (Schneider et al., 2015) of the atoms computed using RDKit; and two variants of *breadth-first search (BFS)*, where we choose the first atom either randomly or as the first atom in canonical order, and then explore the remaining atoms in BFS order, breaking ties between equidistant next nodes randomly.

## 2.4 TRAINING MOLER

MoLeR is trained in the autoencoder paradigm, and so we extend our decoder from above with an encoder that computes a single representation for the entire molecule. This encoder GNN operates directly on the full molecular graph, but is motif-aware through the motif annotations included in the atom features. These annotations are deterministic functions of the input molecule, and thus in principle could be learned by the GNN itself, but we found them to be crucial to achieve good performance. Our model is agnostic to the concrete GNN type; in practice, we use a simple yet expressive GNN-MLP layer, which computes messages for each edge by passing the states of its endpoints through an MLP. Similar to Brockschmidt (2020), we found that this approach outperforms commonly used GNN layers such as GCN (Kipf & Welling, 2016) or GIN (Xu et al., 2018).

We train our overall model to optimize a standard VAE loss (Kingma & Welling, 2013) with several minor modifications, resulting in the linear combination $\lambda_{prior} \cdot \mathcal{L}_{prior}(x) + \mathcal{L}_{rec}(x) + \lambda_{prop} \cdot \mathcal{L}_{prop}(x)$. The weights $\lambda_{prior}$ and $\lambda_{prop}$ are hyperparameters that we tuned empirically. We now elaborate on each of these loss components.

We define $\mathcal{L}_{prior}(x) = -\mathcal{D}_{KL}(q_\theta(z \mid x) || p(z))$, where $p(z)$ is a multivariate Gaussian; as discussed above, the encoder $q_\theta$ is implemented as a GNN followed by two heads used to parameterize the mean and the standard deviation of the latent code $z$. We found that choosing $\lambda_{prior} < 1$ and using a sigmoid annealing schedule (Bowman et al., 2016) was required to make the training stable.

Following our decoder definition above, the reconstruction term $\mathcal{L}_{rec}$ could be written as a sum over the log probabilities of each step $s_i$, conditioned on the partial molecule $M_i$. However, we instead rewrite this term as an expectation with the step chosen uniformly over the entire generation:

$$\mathcal{L}_{rec}(x) = \mathbb{E}_{z \sim q_\theta(z|x)} \mathbb{E}_{i \sim \mathcal{U}} \log p(s_i \mid z, M_i).$$

This makes it explicit that different generation steps for a fixed input molecule do not depend on each other. Figure 1 illustrates this visually, as there are no dependencies between the individual steps. We use this to train in parallel on all generation steps at once (i.e., a batch is made up of many steps like (1)-(5) in Figure 1). Additionally, we *subsample* generation steps, i.e., uniformly at random drop some of the generation steps from training, to get a wider variety of molecules within each batch. These enhancements improve training speed and robustness, and are feasible precisely because our model does not depend on the generation history. In Appendix I.2 we show empirically that subsampling leads to faster convergence on several downstream metrics.

Finally, following prior works (Gómez-Bombarelli et al., 2018; Winter et al., 2019a; Li et al., 2021), we use $\mathcal{L}_{prop}(x)$ to ensure that simple chemical properties can be accurately predicted from the latent encoding of a molecule. Concretely, we use an MLP regressor on top of the sampled latent code $z$ to predict molecular weight, synthetic accessibility (SA) score, and octanol-water partition coefficient (logP), using MSE on these values as objective. We found that choosing the weight $\lambda_{prop}$ of this objective to be smaller than $0.1$ was necessary to avoid the decoder ignoring the latent code $z$. All of these properties can be readily computed from the input molecule $x$ using the RDKit library, and hence do not require additional annotations in the training data. Note that due to the inherent stochasticity in the VAE encoding process, obtaining a low value of $\mathcal{L}_{prop}$ is only possible if the latent space learned by $q_\theta$ is smooth with respect to the predicted properties.

## 3 EXPERIMENTS

**Setup**  We use training data from GuacaMol (Brown et al., 2019), which released a curated set of $\approx$1.5M drug-like molecules, divided into train, validation and test sets. We train MoLeR on the GuacaMol training set until loss on the validation set does not improve; we then use the best checkpoint selected based on validation loss to evaluate on downstream tasks. As discussed above, we found that subsampling generation sequence steps to use only half of the steps per molecule tends to speed up convergence, as it yields more variety within each batch. Therefore, we subsample generation steps for all MoLeR experiments unless noted otherwise. For molecular optimization, we pair MoLeR with Molecular Swarm Optimization (MSO) (Winter et al., 2019b), which is a black-box latent space optimization method that was shown to achieve state-of-the-art performance. For more details on the training routine, experimental setup, and hyperparameters, see Appendix C. We show samples from the model's prior in Appendix D.

**Baselines**  As baselines, we consider three established graph-based generative models: CGVAE (Liu et al., 2018), JT-VAE (Jin et al., 2018), and HierVAE (Jin et al., 2020). Since the publicly released code of Liu et al. (2018) does not scale to datasets as large as GuacaMol, we re-implemented CGVAE following the released code to make it more efficient. For JT-VAE, we used the open-source code, but implemented multithreaded decoding, which made sampling 8x faster. For HierVAE, we used the released code with no changes. Due to the high cost of training JT-VAE and HierVAE, we did not tune their hyperparameters and instead used the default values.

### 3.1 QUANTITATIVE RESULTS

Table 1: Training and sampling speed for our model and the baselines on a Tesla K80 GPU.

**Efficiency**  We measure the speed of different models in training and inference, quantified by the number of molecules processed per second. Note that we do *not* subsample generation steps for this comparison, so that every model processes all the steps, even though MoLeR can learn from only a subset of them. We compare

| Model | Train (mol/sec) | Sample (mol/sec) |
|-------|-----------------|-------------------|
| CGVAE | 57.0 | 1.4 |
| JT-VAE | 3.2 | 3.4 |
| HierVAE | 17.0 | 12.3 |
| MoLeR | **95.2** | **34.2** |

these results in Table 1. We see that, thanks to a simpler formulation and parallel training on all generation steps, MoLeR is much faster than all baselines for both training and inference.

**Unconstrained Generation**  Similarly to Brown et al. (2019), we use Frechet ChemNet Distance (FCD) (Preuer et al., 2018) to measure how much sampled molecules resemble those in the training data. We show the results in Figure 2 (left), in which we compare different models and variations of MoLeR trained with different choices of generation order (see Section 2.3) and different motif vocabulary sizes. It shows that MoLeR with a large vocabulary outperforms the baselines substantially,

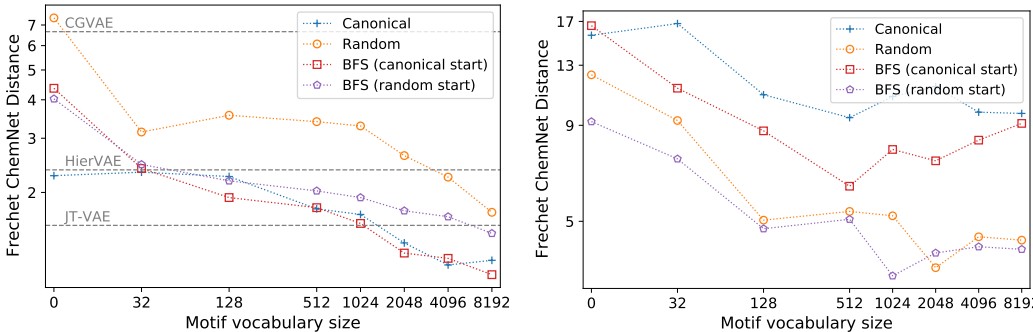

Figure 2: Frechet ChemNet Distance (lower is better) for different generation orders and vocabulary sizes. We consider generation from scratch (left), and generation starting from a scaffold (right).

despite being much faster to train and sample from, and having support for scaffold-constrained generation. Furthermore, we can see that MoLeR's performance increases as the vocabulary size grows. Finally, we note that training with generation orders with a deterministic starting point performs best, and that random order performs less well, as modeling a wide range of orders is harder.

Unlike some prior work (De Cao & Kipf, 2018; Brown et al., 2019), we do not compare validity, uniqueness and novelty, as our models get near-perfect results on these metrics, making comparison meaningless. Concretely, we obtain 100% validity by design (due to the use of valence checks), uniqueness above 99%, and novelty above 97%.

**Scaffold-constrained Generation**    Next, we consider the setting of enforcing a given scaffold. We first choose a chemically relevant scaffold $\Sigma$ (PubChem CID 12658820) that commonly appears in GuacaMol training data. We then estimate the posterior distribution on latent codes induced by $\Sigma$ by encoding all training molecules that contain it and approximating the result with a Gaussian Mixture Model (GMM) with 50 mixture components. Finally, we draw latent codes from the GMM, decode them starting the generation process from $\Sigma$, and compare the resulting molecules with molecules from the data that contain $\Sigma$. By using samples from the GMM-approximated posterior, as opposed to samples from the prior, we ensure that we use latent codes which are *compatible* with the scaffold $\Sigma$, which we found to dramatically improve the downstream metrics. Intuitively, constraining the decoding restricts the latent codes of output molecules to a manifold defined by the scaffold constraint; using an approximate posterior ensures that the projected samples lie close to that manifold.

In Figure 2 (right) we show the resulting FCD. We find that the relative performance of different generation orders is largely *reversed*: since the models trained with canonical order can only complete prefixes of that order, they are not well equipped to complete arbitrary scaffolds. On the other hand, models trained with randomized orders are more flexible and handle the task well. As with generation from scratch, using a larger motif vocabulary tends to help, especially if motifs happen to decompose the scaffold into smaller fragments (or even the entire scaffold may appear in the vocabulary). Finally, we note that BFS order using a random starting point gives the best results for this task, while still showing good performance for unconstrained sampling.

**Unconstrained Optimization**    We experiment on the GuacaMol optimization benchmarks (Brown et al., 2019), tracking two metrics: raw performance score, and quality, defined as absence of undesirable substructures. In Table 2 (left), we compare our results with those taken from the literature. We find that MoLeR maintains a good balance between raw score and quality. Note that the quality filters are not directly available to the models during optimization, and rather are evaluated post-hoc on the optimized molecules. This ensures that high quality scores can only be achieved if the model is biased towards reasonable molecules, and not by learning to exploit and "slip through" the quality filters, similarly to what has been shown for property predictors (Renz et al., 2020). Consequently, the best performing models often produce unreasonable molecules (Winter et al., 2019b; Xu et al., 2020). While the SMILES LSTM baseline of Brown et al. (2019) also gets good results on both score and quality, as we will see below, it struggles to complete arbitrary scaffolds. Note that, out of 20 tasks in this suite, only one tests optimization from a scaffold, and that task uses a small scaffold (Figure 3 (top)), making it relatively easy (even simple models get near-perfect results). In contrast, scaffolds typically used in drug discovery are much more complex (Schuffenhauer et al., 2007; Schuffenhauer, 2012). We conclude that while MoLeR shows good performance on GuacaMol tasks, they do not properly evaluate the ability to complete realistic scaffolds.

Table 2: Results on 20 GuacaMol tasks (left) and 4 additional scaffold-based tasks (right). First five rows correspond to baselines from Brown et al. (2019). We do not compute quality if less than 100 molecules per benchmark were found.

|                | GuacaMol | | Scaffolds | |
| --- | --- | --- | --- | --- |
| Method | Score | Quality | Score | Quality |
| Best of dataset | 0.61 | 0.77 | 0.17 | - |
| SMILES LSTM | 0.87 | 0.77 | 0.45 | - |
| SMILES GA | 0.72 | 0.36 | 0.45 | - |
| GRAPH MCTS | 0.45 | 0.22 | 0.20 | - |
| GRAPH GA | 0.90 | 0.40 | 0.79 | - |
| CDDD + MSO | 0.90 | 0.58 | 0.92 | 0.59 |
| MNCE-RL | 0.92 | 0.54 | 0.95 | 0.47 |
| MoLeR + MSO | 0.82 | 0.75 | 0.93 | 0.63 |

Figure 3: Scaffold from a GuacaMol benchmark (top) and a scaffold from our additional benchmark (bottom).

**Scaffold-constrained Optimization** To evaluate scaffold-constrained optimization, we extend the GuacaMol benchmarks with 4 new scaffold-based tasks, using larger scaffolds extracted from or inspired by clinical candidate molecules or marketed drugs, which are more representative of real-world drug discovery (e.g. Figure 3 (bottom)). The task is to perform scaffold-constrained exploration towards a target property profile; as the components of the scoring functions are aggregated via the geometric mean, and presence of the scaffold is binary, molecules that do not contain the scaffold receive a total score of 0 (see Appendix E for more details). We show the results in Table 2 (right). We see that MoLeR performs well, while most baseline approaches struggle to maintain the scaffold.

Finally, we run the tasks of Lim et al. (2019), where the aim is to generate 100 distinct decorations of large scaffolds to match one or several property targets: molecular weight, logp and TPSA. While the model of Lim et al. (2019) is specially designed to produce samples conditioned on the values of these three properties in one-shot, we convert the target property values into a single objective that we can optimize with MSO. Concretely, for each property we compute the absolute difference to the target value, which we divide by the result of Lim et al. (2019) for a given task, and then average over all properties of interest; under the resulting metric, the model of Lim et al. (2019) gets a score of 1.0 by design. We show the results in Figure 4. Despite not being trained for this this task, MoLeR outperforms the baseline on all benchmarks. These results show that MoLeR can match the capabilities of existing scaffold-based models out-of-the-box. To verify that this cannot be attributed solely to using MSO, we also tried using CDDD as the generative model. We find that CDDD often produces invalid molecules or molecules that do not contain the scaffold; MoLeR avoids both of these problems thanks to valence constraints and scaffold-constrained generation. As a result, in some cases CDDD needs many steps to discover 100 distinct decorations or does not discover them at all; having 100 decorations is needed to compare the average result to Lim et al. (2019). Thus, for CDDD we plot a modified score by duplicating the worst score an appropriate number of times; this was not needed for MoLeR, as it always finds enough decorations in the first few steps.

## 3.2 QUALITATIVE RESULTS

**Unconstrained and constrained interpolation** To test the smoothness of our latent space and analyze how adding the scaffold constraint impacts decoding, we select a chemically relevant scaffold (PubChem CID 7375) and two dissimilar molecules $m_1$ and $m_2$ that contain it. We then linearly interpolate between the latent encodings of $m_1$ and $m_2$, and select those intermediate points at which the corresponding decoded molecule changes. We show in Figure 5 (top) that MoLeR correctly identifies a smooth transition from $m_1$ to $m_2$. Most of the differences between $m_1$ and $m_2$ stem from the latter containing two additional rings, and we see that rings are consistently added during the interpolation. For example, in the third step, the molecule grows by one extra ring, but of a type that does not appear in $m_2$; in the next step, this ring transforms into the correct type, and is then present in all subsequent steps (a similar pattern can be observed for the other ring). However, we see that some intermediate molecules do not contain the scaffold: although all of the scaffold's building blocks are present, the interpolation goes through a region in which the model decides to move the NH group to a different location, thus breaking the integrity of the scaffold. This shows that while latent space distance strongly correlates with structural similarity, latent space smoothness alone does not guarantee the presence of a scaffold, which necessitates scaffold-based generation.

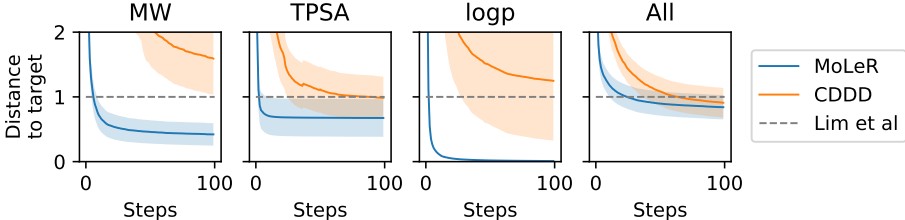

Figure 4: Comparison on tasks from Lim et al. (2019). We show both single-property optimization tasks as well as one where all properties must be optimized simultaneously. We plot averages and standard error over 20 runs for each task; each run uses a different scaffold and property targets.

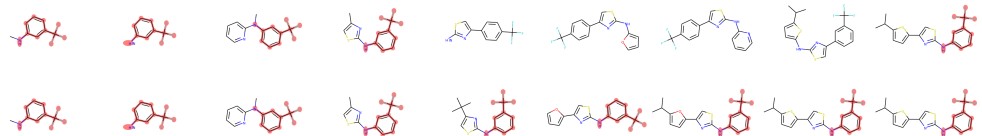

Figure 5: Interpolation between latent encodings of two molecules; unconstrained decoding (top), constrained with scaffold (bottom). The scaffold is highlighted in each molecule that contains it.

In contrast, in Figure 5 (bottom), we show the same sequence of latent codes decoded with a scaffold constraint. We see that the constraint keeps the NH group locked in place, so that all intermediate molecules contain the scaffold. The molecule decoded under a scaffold constraint is typically very similar to the one decoded without, showing that constrained decoding preserves chemical features that are not related to the presence of the scaffold. However, when trying to include the scaffold, our model does not have to resort to a simple rearrangement of existing building blocks: for example, in the 7th step, adding a constraint also modifies one of the ring types, which results in a smoother interpolation in comparison to the unconstrained case. Finally, while the interpolation points were chosen to remove duplicates from the unconstrained path, we see that the last two latent points both get mapped to $m_2$ when the constraint is introduced. This is because the last step of the unconstrained interpolation merely rearranges the motifs, moving back the NH group to its initial location, and restoring the scaffold. This modification is not needed if the scaffold is already present, and therefore our model chooses to project both latent codes to the same point on the scaffold-constrained manifold.

**Latent Space Neighborhood** To analyze the structure of our scaffold-constrained latent space, we select another scaffold and perform scaffold-constrained decoding of a group of neighboring latent points. We note that close-by latent codes decode to similar molecules, often composing the same motifs in a different way, mirroring the observation of Jin et al. (2018). Moreover, some latent space directions seem to track simple chemical properties. For further analysis of this result see Appendix F.

**Learned Motif Representations** To better understand how MoLeR uses motifs, we extract learned motif representations from a trained model. We found that, despite the weights having no *direct* access to molecular structure or features of motifs, nearest neighbors in the representation space correspond to pairs of nearly identical motifs. See Appendix G for visualization and further discussion.

### 3.3 ABLATIONS

We tested a number of ablations of our model to analyze the effect of its individual components. Figure 2 shows the effect of motif vocabulary size; in particular, the points for 0 motifs correspond to not using any motif information at all. In Appendix H we repeat the analysis from Figure 2 for optimization performance, showing that some of the trends transfer also to that setting. Finally, we considered partially removing motifs by not using motif embeddings in the encoder, and found that this also decreases performance, as the model needs to use some of its capacity to recognize motifs in the input graphs (see Appendix I.1 for details).

We also evaluated the impact of subsampling generation steps (as described in Section 2.4), and found that this method allows MoLeR to match the training data more rapidly, as measured by FCD (see Appendix I.2 for details). Finally, we analyzed the influence of using the auxiliary loss term $\mathcal{L}_{prop}$, and found that in a model trained without this objective, molecules that are close in latent space are less similar to each other in important chemical properties such as synthetic accessibility (details can be found in Appendix I.3).

## 4 RELATED WORK

Our work naturally relates to the rich family of *in silico* drug discovery methods. However, it is most related to works that perform iterative generation of molecular graphs, works that employ fragments or motifs, and works that explicitly consider scaffolds.

**Iterative generation of molecular graphs** Many graph-based models for molecule generation employ some form of iterative decoding. Often a single arbitrary ordering is chosen: Liu et al. (2018) first generate all atoms in a one-shot manner, and then generate bonds in BFS order; Jin et al. (2018; 2020) generate a coarsened tree-structured form of the molecular graph in a deterministic DFS order; and You et al. (2018) use random order. Some works go beyond a single ordering: Liao et al. (2019) marginalize over several orders, while Mercado et al. (2020) try both random and canonical, and find the latter produces better samples, which is consistent with our unconstrained generation results. Sacha et al. (2020) generate graph edits with the goal of modeling reactions, and evaluate a range of editing orders. Although the task in their work is different, the results are surprisingly close to ours: a fully random order performs badly, and the optimal amount of non-determinism is task-dependent.

**Motif extraction** Several other works make use of motif extraction approaches related to the one described in Section 2.1. Jin et al. (2020) propose a very similar strategy, but additionally do not break *leaf bonds*, i.e. bonds incident to an atom of degree 1, which we found produces many motifs that are variations of the same underlying structure (e.g. a ring) with different combinations of leaf atoms; for simplicity, we chose to omit that rule in our extraction strategy. More complex molecular fragmentation approaches also exist (Degen et al., 2008), and we plan to explore them in future work.

**Motif-based generation** Our work is closely related to the work of Jin et al. (2018; 2020), which also uses motifs to generate molecular graphs. However, these works cannot be easily extended to scaffold-based generation, and cannot generate molecules which use building blocks not covered by the motif vocabulary. While HierVAE (Jin et al., 2020) does include individual atoms and bonds in its vocabulary, motifs are still assembled in a tree-like manner, meaning that the model cannot generate an arbitrary cyclic structure if its base cycles are not present in the vocabulary.

**Scaffold-conditioned generation** Some prior works can construct molecules under a hard scaffold constraint. Lim et al. (2019) proposes a graph-based model with persistent state; scaffold-based generation is possible because the model is explicitly trained on (scaffold, molecule) pairs. In contrast, MoLeR treats *every* intermediate partial graph as if it were a scaffold to be completed. Li et al. (2018a) use a soft constraint, where the scaffold is part of the input, but is not guaranteed to be present in the generated molecule. Finally, Arús-Pous et al. (2020); Langevin et al. (2020) adapt SMILES-based models to work with scaffolds, and cannot guarantee validity of generated molecules.

Overall, it is non-trivial to extend existing molecule generators to the scaffold setting. For SMILES-based models, the scaffold may not be represented by a substring of the SMILES representation of the complete molecule, and so its presence cannot easily be enforced by forcing some of the decoder's choices. Junction tree-based models (Jin et al., 2018) start the generation at a leaf of the junction tree, but the junction tree for a scaffold does not necessarily match that of the full molecule, or reaches its leaves. Furthermore, graph-based models often use a recurrent state updated throughout the entire generative procedure (Jin et al., 2018; 2020). Finally, fragment-based models that do not support substructures not covered by their fragment library (Jin et al., 2020) fail when a scaffold is not fully covered by known fragments. MoLeR, through seemingly minor design choices (graph-based, conditioned only on a partial graph, not reliant on a fixed generation order, flexibility to switch between fragment-based and atom-by-atom generation), side-steps all of these issues.

## 5 CONCLUSION

In this work, we presented MoLeR: a novel graph-based model for molecular generation. As our model does not depend on history, it can complete arbitrary scaffolds, while still outperforming state-of-the-art graph-based generative models in unconstrained generation. Our quantitative and qualitative results show that MoLeR retains desirable properties of generative models - such as smooth interpolation - while respecting the scaffold constraint. Finally, we show that it exhibits good performance in unconstrained optimization, while excelling in scaffold-constrained optimization.

## 6 ETHICS STATEMENT

As we consider our work to be fundamental research, there are no direct ethical risks or societal consequences; these have to be analyzed per concrete applications. Broadly speaking, tools such as MoLeR can be widely beneficial for the pharmaceutical industry. However, note that while MoLeR performs optimization tasks that would otherwise be done manually by medicinal chemists, it is unlikely to be considered a replacement for them, and rather an enhancement to their creative process.

## 7 ACKNOWLEDGMENTS

We would like to thank Hubert Misztela, Michał Pikusa and William Jose Godinez Navarro for work on the JT-VAE baseline. Moreover, we want to acknowledge the larger team (Ashok Thillaisundaram, Jessica Lanini, Megan Stanley, Nikolas Fechner, Paweł Czyż, Richard Lewis, Sarah Lewis and Qurrat Ul Ain) for engaging in helpful discussions.

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

## A    BACKGROUND: GRAPH NEURAL NETWORKS

In this work we consider graphs $\mathcal{G} = (\mathcal{V}, \mathcal{E})$ with vertices $\mathcal{V}$ and edges $\mathcal{E}$. In the case of molecules, edges $\mathcal{E}$ correspond to bonds between pairs of atoms, and are thus *typed*, with each bond being either single, double or triple. Formally

$$\mathcal{E} \subseteq \mathcal{V} \times \{\mathsf{single}, \mathsf{double}, \mathsf{triple}\} \times \mathcal{V}$$

Note that while cheminformatics tools such as RDKit also distinguish a fourth type of bonds, called *aromatic*, generating aromatic rings in a step-by-step fashion is known to be challenging (Jin et al., 2018). Thus, during preprocessing we convert aromatic rings to alternating single and double bonds, following a process called *kekulization*.

Given $\mathcal{G}$, we learn node and graph representations using Graph Neural Networks. The network begins with starting node representations $\{h_v^0 : v \in \mathcal{V}\}$; in our case, we set these to linear projections of the node features $h_v^{(init)}$. Each GNN layer propagates node representations $\{h_v^t : v \in \mathcal{V}\}$ to compute $\{h_v^{t+1} : v \in \mathcal{V}\}$ using message passing (Gilmer et al., 2017):

$$h_v^{t+1} = f(h_v^t, \mathsf{aggregate}(\{m_\ell(h_v^t, h_u^t) : (v, \ell, u) \in \mathcal{E}\}))$$

where $m_\ell$ computes the message between two nodes connected by an edge of type $\ell$, aggregate combines all messages received by a given node, and $f$ computes the new node representation given the old representation and the aggregated messages. A common choice is to use a linear layer for every $m_l$, a pointwise sum for aggregate, and a GRU update for $f$ (Li et al., 2015), but many other variants exist (Brockschmidt, 2020; Corso et al., 2020).

After $L$ layers of message passing we obtain final representations $\{h_v^L : v \in \mathcal{V}\}$, with $h_v^L$ summarizing the $L$-hop neighborhood of $v$. These representations can be pooled to form a graph-level representation by using any permutation-invariant aggregator, such as a weighted sum.

## B    ARCHITECTURE

The backbone of our architecture consists of two GNNs: one used to encode the input molecule, and the other used to encode the current partial graph. Both GNNs have the same architecture, but are otherwise completely separate, and do not share any parameters.

To implement our GNNs, we employ the GNN-MLP layer (Brockschmidt, 2020). We use 12 layers with separate parameters, Leaky ReLU non-linearities (Maas et al., 2013), and LayerNorm (Ba et al., 2016) after every GNN layer. If using motifs, we concatenate the atom features with a motif embedding of size 64, and then linearly project the result back into 64 dimensions. We use 64 as the hidden dimension throughout all GNN layers, guided by early experiments showing that wider hidden representations were less beneficial than a deeper GNN. Moreover, to improve the flow of gradients in the GNNs, we produce the final node-level feature vectors by concatenating both initial and intermediate node representations across all layers, resulting in feature vectors of size $64 \cdot 13 = 832$. Intuitively, this concatenation serves as a skip connection that shortens the path from the node features to the final representation.

To pool node-level representations into a graph-level representation, we use an expressive multi-headed aggregation scheme. The $i$-th aggregation head consists of two MLPs: $s^i$ which computes a scalar aggregation score, and $t^i$ which computes a transformed version of the node representation. These are then used to compute the $i$-th graph-level output $o^i$ according to

$$
\begin{aligned}
w^i &= \mathsf{normalize}(\{s^i(h_v) : v \in \mathcal{V}\}) \\
o^i &= \sum_{v \in \mathcal{V}} w_v^i \cdot t^i(h_v)
\end{aligned}
$$

Specifically, we compute the scores $s^i(h_v)$ for all of the nodes, normalize across the graph using normalize, and then use them to construct a weighted sum of the transformed representations $t^i(h_v)$. For the normalization function we consider either passing the scores through a softmax (which results in a head that implements a weighted mean) or a sigmoid (weighted sum). We use 32 heads for the encoder GNN, and 16 heads for the partial graphs GNN. In both cases, half of the heads use a softmax normalization, while the other half uses sigmoid. The outputs from all heads are concatenated to form the final graph-level vector; as different heads use different normalization functions (softmax or sigmoid), this is in spirit related to Principal Neighborhood Aggregation (Corso et al., 2020), but here used for graph-level readout instead of aggregating node-level messages.

Our node aggregation layer allows to construct a powerful graph-level representation; its dimensionality can be adjusted by varying the number of heads and the output dimension of the transformations $t_i$. For input graphs we use a 512-dimensional graph-level representation (which is then transformed to produce the mean and standard deviation of a 512-dimensional latent code $z$), and for partial graphs we use 256 dimensions.

To implement the functions used in our decoder procedure (i.e., the neural networks implementing PickAtomOrMotif, PickAttachment, and PickBond in Algorithm 1), we use simple multilayer perceptrons (MLPs).

The MLP for PickAtomOrMotif has to output a distribution over all atom and motif types and the special END_GEN option. As input, it receives the latent code $z$ and the partial molecule representation $h_{mol}$. As the number of choices is large, we use hidden layers which maintain high dimensionality (two hidden layers with dimension 256). Predicting the type of the first node in an empty graph would require encoding an *empty* partial molecule to obtain $h_{mol}$; in practice, we side-step this technicality by using a separate MLP to predict the first node type, which takes as input only the latent encoding $z$.

In contrast, the networks for PickAttachment and PickBond are used as scorers (i.e. need to output a single value), therefore we use MLPs with hidden layers that gradually reduce dimensionality (concretely, three hidden layers with dimension 128, 64, 32, respectively). The MLPs for PickAttachment and PickBond take the latent code $z$, the partial molecule representation $h_{mol}$, and the representation $h_v$ of each scored candidate node $v$. Finally, PickBond not only needs to predict the partner of a bond, but also one of three bond types (single, double and triple); for that we use an additional MLP with the same architecture as the scoring network, but used for classification.

## C  TRAINING AND INFERENCE

We train our model using the Adam optimizer (Kingma & Ba, 2014). We found that adding an initial warm-up phase for the KL loss coefficient $\lambda_{prior}$ (i.e. increasing it from 0 to a target value over the course of training) helps to stabilize the model. However, our warm-up phase is relatively short: we reach the target $\lambda_{prior}$ in 5000 training steps, whereas full convergence requires around 200 000 steps. This is in contrast to Jin et al. (2018), which varies $\lambda_{prior}$ (also referred to as $\beta$) uniformly over the entire training. A short warm-up phase is beneficial, as it allows to perform early stopping based on reaching a plateau in validation loss; this cannot be done while $\lambda_{prior}$ is being varied, as there is no clear notion of improvement if the training objective is changing.

When constructing minibatches during training, we combine the molecular graphs until a limit of 25 000 nodes is reached. We cap the total number of nodes rather than the total number of molecules, as that is more robust to varying sizes of molecules in the training data.

### C.1  HYPERPARAMETER TUNING

Due to a very large design space of GNNs, we performed only limited hyperparameter tuning during preliminary experiments. In our experience, improving the modeling (e.g. changing the motif vocabulary or generation order) tends to have a larger impact than tuning low-level GNN architectural choices. For hyperparameters describing the expressiveness of the model, such as the number of layers or hidden representation size, we set them to reasonably high values, which is feasible as our model is very efficient to train. We did not make an attempt to reduce model size; it is likely that a smaller model would give equivalent downstream performance.

One parameter that we found to be tricky to tune is the $\lambda_{prior}$ coefficient that weighs the $\mathcal{L}_{prior}$ loss term. An additional complication stems from the fact that we compute $\mathcal{L}_{rec}$ as an average over the generation steps instead of a sum. While we made this design choice to make the loss scaling robust to training steps subsampling (i.e. $\lambda_{prior}$ does not have to be adjusted if we use only a subset of steps at training time), it led to decreased robustness when *the difficulty of an average step* varies between experiments. Concretely, when a larger motif vocabulary is used, generating a molecule entails fewer steps, but those steps are harder on average, since the underlying classification tasks distinguish between more classes. In preliminary experiments, we noticed the optimal value of $\lambda_{prior}$ increased with vocabulary size, closely following a logarithmic trend: doubling the motif vocabulary size translated to the optimum $\lambda_{prior}$ increasing by $0.005$. For vocabulary sizes up to 32 we used $\lambda_{prior} = 0.01$, and then followed the logarithmic trend described here. Note that, due to differences in the loss definitions, our value of $\lambda_{prior}$ is not directly comparable to the values for $\beta$ in $\beta$-VAE works.

Finally, varying the generation order and the size of the motif vocabulary explores different performance trade-offs depending on the downstream task. For all optimization benchmarks in Section 3 we used $128$ motifs; moreover, we chose the BFS order with a random starting point for unconstrained optimization, and the fully random order for scaffold-constrained optimization.

## C.2 SOFTWARE AND HARDWARE

We performed all experiments on a single GPU. For all measurements in Table 1, we used a machine with a single Tesla K80 GPU. Our own implementations (MoLeR, CGVAE) are based on TensorFlow 2 (Abadi et al., 2016), while the models of Jin et al. (2018; 2020) (JT-VAE, HierVAE) use PyTorch (Paszke et al., 2019).

Training MoLeR requires first preprocessing the data, which takes up to one CPU day for GuacaMol, followed by training itself, which takes up to a few GPU days. While the generation benchmarks are cheap to run, optimization benchmarks are typically expensive. Each individual optimization benchmark takes between 6 and 130 hours of GPU time, depending on the details of the scoring function and size of the molecules that the algorithm ends up exploring. In particular, scaffold-based optimization benchmarks on average tend to be more compute intensive, as for full correctness the scoring functions need to verify that the scaffold is present (even though with MoLeR it is guaranteed to be included).

## C.3 OPTIMIZATION

To perform optimization we used the original MSO code of Winter et al. (2019b); we found that the default hyperparameters already resulted in good performance. However, we made two modifications to the algorithms to make the interplay of MSO and MoLeR smoother.

**Deterministic encoding**  Despite being a black-box optimization method, MSO does use the *encoder* part of the generative model: first, to encode the seed molecules, but more interestingly, to re-encode molecules found in each step of optimization, adjusting the particle positions as $x \leftarrow \text{encode}(\text{decode}(x))$; we hypothesise that the latter was introduced to "snap back" the particles to the latent space region "preferred" by the encoder. Unlike CDDD, MoLeR is a *variational* autoencoder, thus by design the encoding process is non-deterministic; this randomness interacts badly with MSO's re-encoding. Therefore, for all of our optimization experiments we made the MoLeR encoder deterministic by always returning the maximum likelihood latent code $z$ (which coincides with the mean of the predicted Gaussian).

**Latent code clipping**  One detail of MSO that we adapted to MoLeR is clipping of the particles' latent coordinates. Winter et al. (2019b) clip to a hypercube $[-1, 1]^D$ where $D = 512$ is the latent space dimension; while this makes sense for an unregularized autoencoder such as CDDD, the output of MoLeR's encoder is regularized through the $\mathcal{L}_{prior}$ loss term. Concretely, the mean of the distribution predicted by the encoder is penalized proportionally to its *norm*. This suggests that a ball may better approximate the encoder's distribution than a hypercube, which we indeed found to hold in practice. Therefore, for MoLeR we clip to a ball of fixed radius $R = 10$; on the GuacaMol benchmarks (Brown et al., 2019) this modification alone improved MoLeR's score from $0.77$ to $0.82$, while also improving quality from $0.74$ to $0.76$. We chose the radius $R$ so that *almost all* encodings of training set molecules land within the corresponding ball.

# D   SAMPLES FROM THE PRIOR

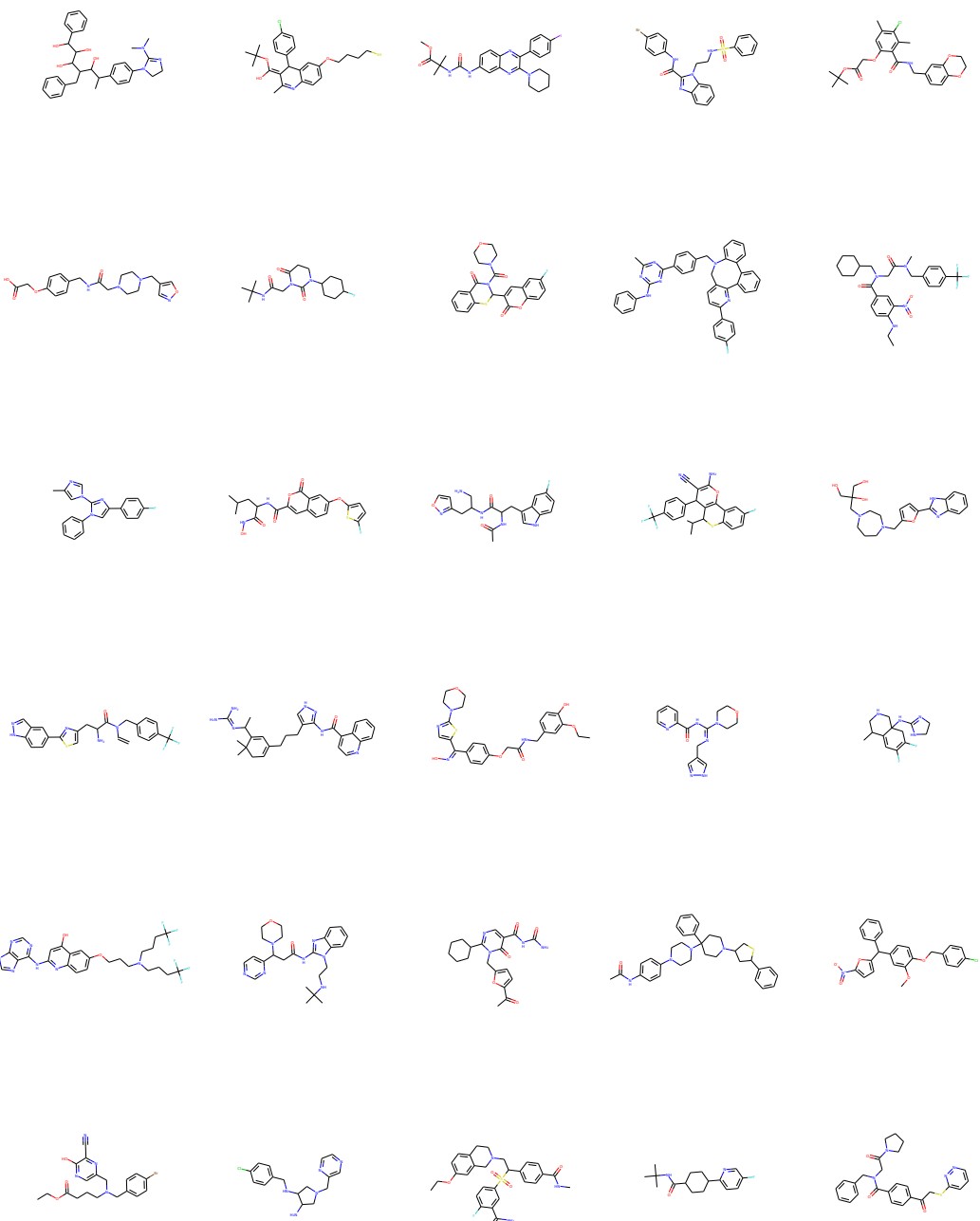

Figure 6: Samples from the prior of a trained MoLeR model.

# E    SCAFFOLD-BASED OPTIMIZATION BENCHMARKS

Our new scaffold-based benchmarks were inspired by real-world clinical candidates or marketed drugs, and employ large challenging scaffolds. The format closely follows the one used for tasks in Guacamol (Brown et al., 2019); in all cases, the score is a task-specific real number in the $[0, 1]$ range, with higher values being better, indicating how well the given molecules match a target molecular profile. To measure quality, we used the same quality filters as Brown et al. (2019).

Table 3: Targets and scaffolds used in our new scaffold-based optimization benchmarks.

| Name | Scaffold | Similarity target | Property profile target |
|---|---|---|---|
| pde5 | Modified scaffold from Sildenafil | Sildenafil | - |
| lorlati | Scaffold inspired by Lorlatinib | Inspired by Lorlatinib's macrocyclization strategy[1] | - |
| GCCA1 | Modified scaffold from Lenacapavir | Inspired by Lenacapavir | - |
| factor xa | Scaffold from Apixaban | Xarelto | Apixaban |

We designed four tasks, three of which ask to maximize similarity towards a fixed target molecule (which may already have some of the required properties we care about in a drug discovery project

---

[1]Pyrazol and phenyl have been exchanged to reduce similarity to the original molecule.

e.g. binding), while enforcing the presence of a scaffold (which is not present in the target, making the task non-trivial). In real-world drug design, this scenario is known as scaffold hopping. Finally, the fourth task, apart from a scaffold, uses two target molecules, maximizing structural similarity to one (Xarelto), while maintaining the properties of the other (Apixaban). In Table 3 we show all molecules used to define our tasks, along with references to the PubChem database. The target molecules have been inspired by existing drugs, however, to make the tasks harder, have in most cases been structurally modified such that they are not present in the Guacamol dataset, which is used by some of the algorithms (e.g. GraphGA, SMILES LSTM) to select the set of starting molecules.

## F   LATENT SPACE NEIGHBORHOOD

In this section, we present more details on the latent space neighborhood learned by MoLeR. For the purpose of this analysis we fixed a scaffold (PubChem CID 57732551), and chose an arbitrary molecule $m$ that contains it. In order to visualize the neighborhood of $m$, we encode it, and then decode a 5 x 5 grid of neighboring latent codes centered at the encoding of $m$. To produce the grid, we choose two random orthogonal directions in the latent space, and then use binary search to select the smallest step size which results in all 25 latent points decoding to distinct molecules. We show the resulting latent neighborhood in Figure 7, where the scaffold is highlighted in each molecule. We see that the model is able to produce reasonable variations of $m$, while maintaining local smoothness, as most adjacent pairs of molecules are very similar. Moreover, we notice that the left-to-right direction is correlated with size, showing that the latent space respects basic chemical properties. If the same 25 latent codes are decoded without the scaffold constraint, only 9 of them end up containing the scaffold, while the other 16 contain similar but different substructures.

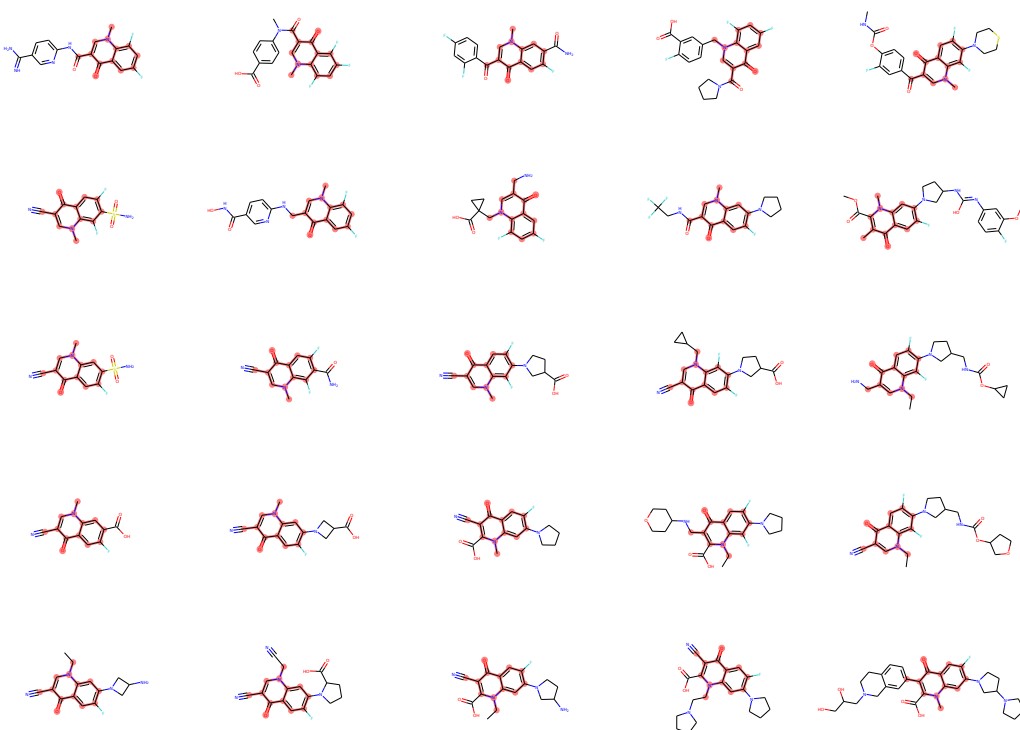

Figure 7: Latent space neighborhood of a fixed molecule containing a chemically relevant scaffold. Each latent code is decoded under a scaffold constraint, so that the desired scaffold (highlighted in red) is present in each molecule.

## G    LEARNED MOTIF REPRESENTATIONS

To extract learned motif representations from a trained MoLeR model, we could use any of its weights that are motif-specific. We first examine the embedding layer in the *encoder*, which is used to construct atom features. Interestingly, we find that these embeddings do not cluster in any way, and even very similar motifs are assigned distant embeddings. We hypothesize that this is due to the use of a simple classification loss function for $\mathcal{L}_{rec}$, which asks to recover the exact motif type, and does not give a smaller penalty for predicting an incorrect but similar motif. Therefore, for two motifs that are similar on the atom level, it may be beneficial to place their embeddings further apart, since otherwise it would be hard for the GNN to differentiate them at all. We hope this observation can inspire future work to scale to very large motif vocabularies, but use domain knowledge to craft a soft reconstruction loss that respects motif similarity.

Now, we turn to a different set of motif embeddings, which we extract from the last layer of the next node prediction MLP in the *decoder*. This results in one weight vector per every output class (i.e. atom and motif type); in contrast to the encoder-side embeddings, the role of these weight vectors is *prediction* rather than *encoding*. We find that pairs of motif embeddings that have high cosine similarity indeed correspond to very similar motifs, which often differ in very subtle details of the molecular graph. We show some of the closest pairs in Figure 8.

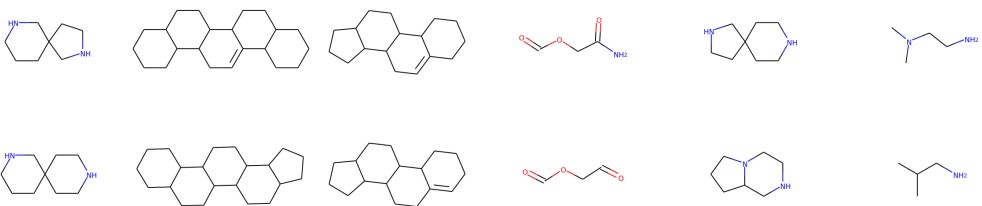

Figure 8: Six pairs of similar motifs (one per column), as extracted from weights of a trained MoLeR model.

Finally, note that the motif embeddings discussed here (both encoder side and decoder side) were trained end-to-end with the rest of the model, and did not have direct access to graph structure or chemical features of motifs. Therefore, there is no bias that would make embeddings of similar motifs close, and this can only arise as a consequence of training.

## H    EFFECT OF USING MOTIFS ON OPTIMIZATION PERFORMANCE

In Figure 2 we show how the choice of generation order and motif vocabulary size impact the samples generated by MoLeR. In Figure 9 we mirror this analysis, looking at optimization performance on both groups of tasks reported in Table 2: original tasks of Brown et al. (2019), and our scaffold-based tasks.

For unconstrained optimization, we see that using motifs generally improves results for most generation orders, but the trends are much more noisy than in the case of generation performance. We speculate this may be caused by the fact that MoLeR is not directly trained for optimization performance, and thus we see more variance between reruns due to randomness.

For scaffold-constrained optimization, we see some improvement when motifs are introduced ($0 \rightarrow 32$), but there is no improvement with larger motif vocabularies, which we attribute to the fact that our scaffold-based tasks use very large scaffolds, and so their optimal decorations typically do not contain large or uncommon motifs. Finally, we note that MoLeR trained under a canonical order performs competitively in unconstrained optimization, but underperforms in scaffold-constrained optimization, matching the insights from Section 3.1.

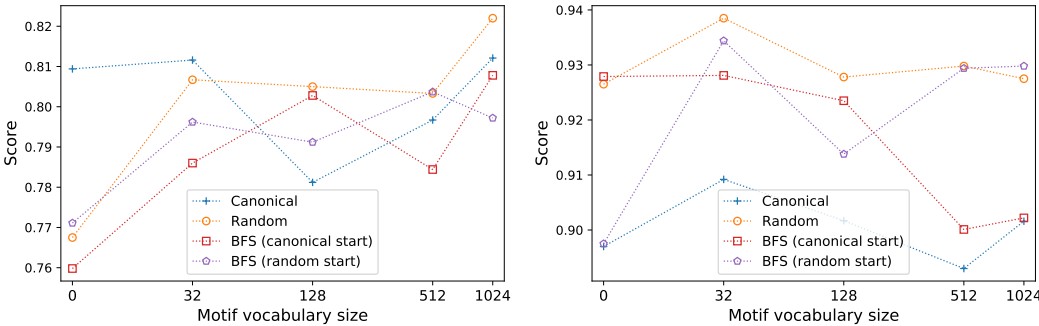

Figure 9: Optimization performance (higher is better) for different generation orders and vocabulary sizes. We separately show the original GuacaMol benchmarks (left), and our new scaffold-based benchmarks (right).

# I  ABLATION STUDIES

In this section, we perform ablation studies to understand the contribution of different design choices to MoLeR's performance.

## I.1  ADDING MOTIF EMBEDDINGS AS INPUT FEATURES

In Section 2.1, we introduced *motif embeddings*, which allow the encoder network to be motif-aware without having to learn to simulate the motif decomposition algorithm. Since the decoder network must reassemble the molecule using the right motifs, it is crucial for the encoder to understand which motifs are present, and including this information explicitly simplifies the learning task.

To confirm this intuition, we ran two MoLeR training runs: one using motif embeddings, and one using only atom-level features. To compare the models, we observed how samples from the prior evolved over the course of training. Concretely, every $5\,000$ training steps we drew $10\,000$ samples from the prior of each model, and computed three metrics using the GuacaMol package (Brown et al., 2019): uniqueness (defined as a fraction of unique samples; higher is better), KL divergence to the training set (defined over several simple chemical properties and then transformed into the $[0, 1]$ range; higher is better) and Frechet ChemNet Distance to the training set (defined as a divergence in intermediate activations of ChemNet; lower is better). We show the results of this in Figure 10. We see that without motif embeddings, MoLeR takes longer to learn to match the training data; this is most pronounced when comparing Frechet ChemNet distance.

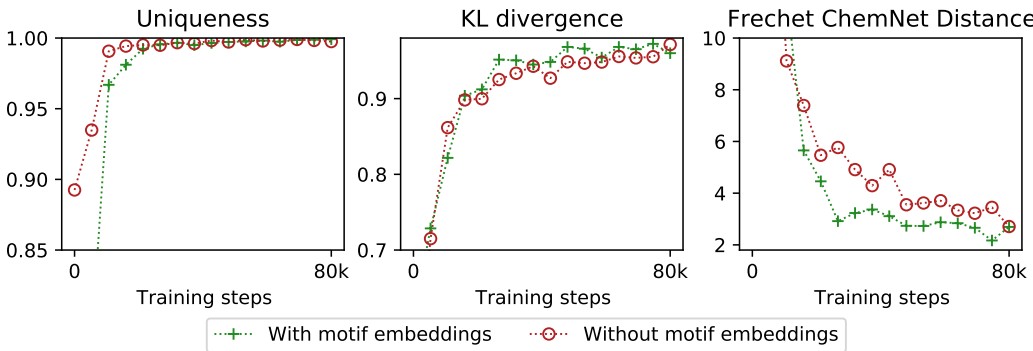

Figure 10: Generation metrics during training, measured for MoLeR both with and without motif embeddings. Using motif embeddings simplifies the learning task, improving the quality of downstream samples.

## I.2 SUBSAMPLING GENERATION STEPS

In Section 2.4, we introduced *generation step subsampling* as a way to get more variety within each training batch. Intuitively, different generation steps can be thought of as separate training samples for the decoder network. As generation steps corresponding to the same full molecule are correlated, subsampling them decreases the intra-batch correlation.

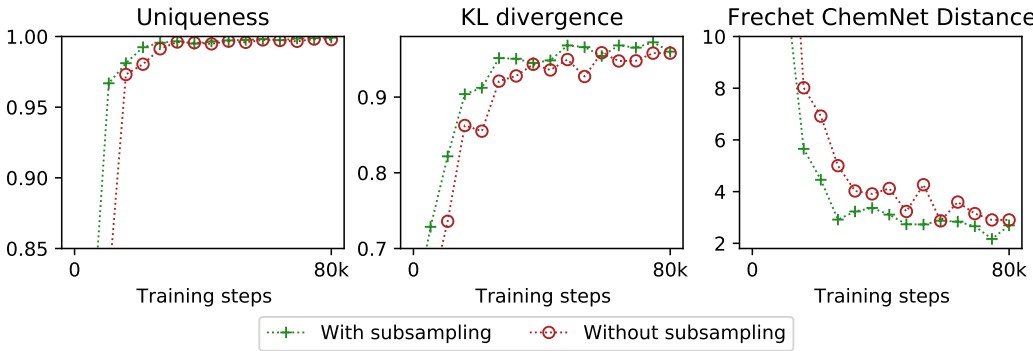

Figure 11: Generation metrics during training, measured for MoLeR both with and without training step subsampling. Subsampling training steps leads to faster convergence on all downstream metrics.

To show that the subsampling strategy is effective in practice, we ran an ablation study, comparing MoLeR trained with and without subsampling, following the same methodology as in Appendix I.1. We show the results in Figure 11. We see that while both runs eventually reach roughly the same performance, the run employing subsampling coverges faster across all metrics; the difference is especially pronounced in the first few thousand training steps.

## I.3 ENCOURAGING LATENT SPACE SMOOTHNESS WITH A PROPERTY PREDICTION LOSS

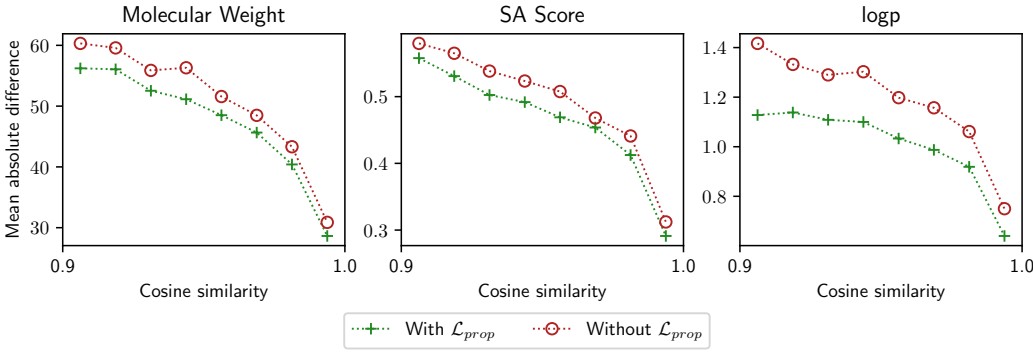

Figure 12: Mean difference in property value when decoding random pairs of latent codes with a given cosine similarity, shown for MoLeR both with and without the $\mathcal{L}_{prop}$ loss. Adding the property prediction loss increases the correlation between latent space distance and property value.

Finally, we analyze the contribution of the $\mathcal{L}_{prop}$ auxiliary loss to the shape of the latent space. In particular, we are interested in understanding whether the latent space is indeed "smooth" with respect to the predicted properties. To quantify this, we consider pairs of latent vectors with cosine similarity in the $(0.9, 1.0)$ range, which we uniformly split into 8 buckets. We continue drawing pairs of samples and adding them to their respective buckets (or discarding them, if the similarity is less than $0.9$), until each bucket has at least $1\,000$ pairs. For each bucket, we compute the mean absolute difference of molecular weight, synthetic accessibility (SA) score, and octanol-water partition coefficient (logP) between the two molecules of each pair and plot this in Figure 12. The results show that using the additional objective indeed makes molecules decoded from very similar latent codes have more similar properties.

