# OpenReview forum: "Learning to Extend Molecular Scaffolds with Structural Motifs"
_ICLR.cc/2022/Conference — ICLR 2022 Poster_

### Official Review · Reviewer_9yrY · 2021-10-31

**Correctness:** 3
**Technical Novelty And Significance:** 3
**Empirical Novelty And Significance:** 3
**Recommendation:** 8
**Confidence:** 3

**Main Review:**

Strengths:

This paper proposes a flexible framework that can use motifs to generate molecules, as well as support atom-by-atom generation. Using a scaffold as the initial seed can make sure the generated molecules contain the specified scaffold. The decoder is not conditioned on the generation history and it conditions only on the hidden vector and the current partial molecule. This design strategy makes the proposed method can be trained using all generation steps in the same batch.

Experimental evaluation is comprehensive. The authors conduct extensive experiments in the unconstrained generation, scaffold-constrained generation, unconstrained optimization, and scaffold-constrained optimization to demonstrate the effectiveness of the proposed method. They also have ablation studies to verify the effectiveness of different components. The proposed method performs comparably to baselines on unconstrained molecular optimization tasks and outperforms them on scaffold-based tasks.

Weaknesses:

Baseline methods are not comprehensive. Some important graph-based [1] or fragment-based [2] molecule generations are not included in the experimental comparison. Especially for the training and sampling speed evaluation, the proposed method needs to embed the partial molecule graph at every generation step, which seems inefficient and slow. Are the authors aware of other methods that do not need to embed the partial graph at every step?

Important related works are not included. [3] works on a different problem but also generates molecules using a similar atom-by-atom method. Particularly, [3] has also investigated different strategies to determine the generation order, which is closely related to this work. The authors mention that it is necessary to use a small weight for the KL loss and increase its weight from 0 gradually to make the VAE model training stable. This strategy has been extensively studied in [4] which investigates the posterior collapse problem of VAE for molecule sequence generation. Could the authors include the closely related work [3, 4] for references?



[1] Shi, Chence, et al. "Graphaf: a flow-based autoregressive model for molecular graph generation." arXiv preprint arXiv:2001.09382 (2020).

[2] Podda, Marco, Davide Bacciu, and Alessio Micheli. "A deep generative model for fragment-based molecule generation." International Conference on Artificial Intelligence and Statistics. PMLR, 2020.

[3] Sacha, Mikołaj, et al. "Molecule edit graph attention network: modeling chemical reactions as sequences of graph edits." Journal of Chemical Information and Modeling 61.7 (2021): 3273-3284.

[4] Yan, Chaochao, et al. "Re-balancing variational autoencoder loss for molecule sequence generation." Proceedings of the 11th ACM International Conference on Bioinformatics, Computational Biology and Health Informatics. 2020.


**Summary Of The Paper:**

This paper proposes a graph-based generative model for molecule generation. The proposed framework MoLeR can use scaffolds as the initial seed to incrementlly generate molecules motif by motif or atom by atom so that the generated molecules consist of the specified scaffold. Experiments show the proposed method performs comparably to existing methods on unconstrained molecular optimization tasks and outperforms these methods on scaffold-based tasks. At the same time, the proposed method is more efficient due to that it is not conditioned on the generation history.

**Summary Of The Review:**

The proposed molecule graph generation method is flexible and achieves good experimentals results.

---

> ### Author Response · Authors · 2021-11-11
> **Review response**
>
> Thank you for your review and the positive comments about our work! Below we address each one of your points one by one.
>
> **Q1**: Additional graph-based [R1] and fragment-based [R2] baselines should be compared to.
>
> **A1**: For the comparison on Guacamol, we compared to SotA on that task; we therefore didn't include [R1] as it uses a different dataset and doesn't evaluate on Guacamol. [R2] is not directly applicable to optimisation according to the authors of that work, which is the task we focus on. However, the paper is interesting and relevant, so we will cite it in our work.
>
> **Q2**: The proposed method embeds the partial graph at every generation step, which seems inefficient. Are the authors aware of other methods that do not need to embed the partial graph at every step?
>
> **A2**: Yes; we can think of at least three ways to avoid re-embedding. A method could:
>
> - (a) have no notion of steps at all, and generate molecules in one shot (e.g. [1])
>
> - (b) be auto-regressive, but based on a recurrent state and not the partial molecule (e.g. LSTM-based models, e.g. CDDD [2], though we note that Transformer-based models implicitly re-encode the partial output at each step)
>
> - (c) be auto-regressive, and based on the partial molecule, but employ an incremental GNN method to make recomputing the embedding of the updated graph cheaper
>
> (a) and (b) give up the possibility of using a scaffold as an initial partial graph (we elaborate in the paper on why that's the case for approaches falling under (b)). (c) could be an interesting extension to MoLeR, but we leave that to future work.
>
> Finally, while MoLeR does need to re-encode partial graphs, due to the use of motifs there are less generation steps on average (and thus less calls to the partial graph encoder GNN). In practice, we found that we could optimize towards modern real-world drug discovery objectives using MoLeR+MSO in a matter of a few hours, so the current level of efficiency is enough for practical use. However, we expect that it's possible to improve MoLeR's efficiency further.
>
> **Q3**: [R3] should be cited, as it also investigates different strategies to determine the generation order.
>
> **A3**: Agreed that it's related, in fact, we're already citing it :-) See the "Related Works" section: "Sacha et al. generate graph edits with the goal of modelling reactions, and evaluate a range of editing orders. Although the task in their work is different, the results are surprisingly close to ours: a fully random order performs badly, and the optimal amount of non-determinism is task-dependent.". We note that we cited the arxiv version, not the journal version, but can change this if the latter is better.
>
> **Q4**: The authors mention that it is necessary to use a small weight for the KL loss and increase its weight from 0 gradually to make the VAE model training stable. This strategy has been extensively studied in [R4], so it should be cited.
>
> **A4**: Makes sense; we'll refer to [R4] when we mention the KL loss weight schedule.
>
> **References**
>
> [1] GraphVAE: Towards Generation of Small Graphs Using Variational Autoencoders
>
> [2] Learning Continuous and Data-Driven Molecular Descriptors by Translating Equivalent Chemical Representations
>
>
>
> [R1] GraphAF: a flow-based autoregressive model for molecular graph generation
>
> [R2] A deep generative model for fragment-based molecule generation
>
> [R3] Molecule edit graph attention network: modeling chemical reactions as sequences of graph edits
>
> [R4] Re-balancing variational autoencoder loss for molecule sequence generation

---

> > ### Comment · Reviewer_9yrY · 2021-11-29
> > **Solved my concerns**
> >
> > Thank the authors' response, which solved my concerns. After all the review and author responses, I decided to maintain my score as accept.

---

> > > ### Author Response · Authors · 2021-11-29
> > > **Thank you for your response**
> > >
> > > Thank you, we're glad to hear your concerns were resolved! In light of that, would you consider increasing your confidence score?

---

### Official Review · Reviewer_V5A5 · 2021-11-02

**Correctness:** 3
**Technical Novelty And Significance:** 3
**Empirical Novelty And Significance:** 4
**Recommendation:** 8
**Confidence:** 4

**Main Review:**

Among the strengths , i identify
- that the model is practical and pragmatic. It strikes interesting balance of flexibility vs. first-principles, where some technical innovations are put at the service of the task/
The "multi-hot objective" (pg 3) and  "GMM posterior" (pg 6) are both interesting and have not been applied frequently in molecule generative models. I find both compelling and helpful. I suspect they can also be applicable to other related works in the future.
- scaffold-based generation is a meaningful problem in molecular design and the existing tools are not great. The authors text in page 9 about SMILES, graph and junction-tree models is compelling, and their approach does side step all these issues while substracting computational complexity

On the weakness,
- Still not order invariant. These canonical ordering procedures are a little arbitrary, and "frontier" is degenerate. This is a fundamental issue that cannot be easily addressed, and the authors assess its impacts quite thoroughly. It is interesting, and further proof that permutation invariance is desirable, that the rank in performance in unconstrained generation is opposite to scaffold-constrained generation.
- Pg 5 "we did not tune their hyperparameters and instead used the default values" This is likely an important reason why the baseline models underperform, since it's not an apples to apples comparison
- except for the benchmarks (see above) I don't see any obvious incremental ways to build upon this class of models. I wonder if the authors could provide a roadmap for high-risk high-reward changes they could propose? Or perhaps this is a terminal approach without much obvious room for incremental improvement.


Some questions that come to mind is

Pg 3 "In each step, it first selects a new atom or entire motif to add to the current partial molecule, or to stop the generation." It is possible for the model to select an atom that is incompatible because there are no more active positions to connect to? That is, if the current M does not have any -H (which are the implicit reactive sites) will it know to stop?

Is there a path (and how likely is it to happen) for a motif to be reconstructed atom by atom even if it is in the vocabulary? Are there any such cases in the training data?


**Summary Of The Paper:**

This work proposes a generative model for molecules. The approach uses a library of motifs, extracted from the training data by breaking down molecules through acyclic bonds adjacent to a cycle. The model can also sample individual atoms. The method is set up in an autoencoder fashion with a GCN encoder that learns node and edge embeddings that are projected into a vectorial latent space. The nodes are initialized with features representing which motif they belong to, so the latent space is motif-aware. The generative approach from the latent space is reminiscent of autoregressive models, but it does not marginalize over the full sequential set of steps, and only through single-step transformations. Seeded with a starting atom or motif, the model sequentially samples the motif (or atom) to be connected and which atoms will be connected. Because more than one path can connect an initial and a final product, it is important to train over multiple paths

**Summary Of The Review:**

I find the work overall convincing. It takes a pragmatic approach that uses relatively sophisticated (but not wildly innovative tools) to improve one obvious flaw in motif-based models which is their inability to invent new motifs. I find the addition of per-atom sampling a convincing escape route for that failure mode.

The results are relatively convincing in the innovation is practically useful, although it is hard to quantify the size of the gains, because hyperparameter optimization strategies were different for the new method and the baselines. Overall, I think it is acceptable and I would be interested in using a model like this in the practice.

---

> ### Author Response · Authors · 2021-11-11
> **Review response**
>
> Thank you for your review and the positive comments about our work! Below we address each one of your points one by one.
>
> **Q1**: Still not order invariant. Canonical ordering procedures are arbitrary, and "frontier" is degenerate. This is a fundamental issue that cannot be easily addressed, and the authors assess its impacts quite thoroughly. It is interesting, and further proof that permutation invariance is desirable, that the rank in performance in unconstrained generation is opposite to scaffold-constrained generation.
>
> **A1**: We agree that the canonical ordering (as defined by rdkit) is a rather arbitrary heuristic. On the other hand, MoLeR still gets good results under a fully random generation ordering, where it learns to generate molecules under any valid generation order. Still, one needs to choose *some* generation order. If we assume the latent space has constant size, then we feel it might even be impossible to be fully order invariant (under some extra assumptions, including that the model has to work in polynomial time, and that P != NP).
>
> **Q2**: "We did not tune their hyperparameters and instead used the default values" - this is likely an important reason why the baseline models underperform, since it's not an apples to apples comparison.
>
> **A2**: See "response to all reviewers".
>
> **Q3**: How to build upon this class of models? Could you provide a roadmap for high-risk high-reward changes? Or is this is a terminal approach without much obvious room for incremental improvement?
>
> **A3**: Interesting question! MoLeR is certainly not a terminal approach (as mentioned in "response to all reviewers", we're using it in practice, and plan to improve the model continuously). Here are some directions we're considering:
> - There is a lot of room to make the optimization/generation more controllable by chemists. Simple extensions include restricting how scaffolds are extended (e.g. disallowing some combinations of decoration + scaffold attachment point), or what building blocks are used (e.g. disallowing the use of some motifs, or combinations thereof). In the extreme, we could even move towards an interactive optimization system (where the chemist would work on a high-level, with lower-level tweaks to the molecular structure being proposed by MoLeR).
> - Even though optimization with MSO works well, it's still a black-box optimizer, and we expect results could be improved with an optimization algorithm better suited for MoLeR. In particular, MSO keeps the decoder parameters fixed, but there are alternatives that may be more flexible (e.g. RL, hill climbing).
>
> **Q4**: Is it possible for the model to select an atom that is incompatible because there are no more active positions to connect to?
>
> **A4**: In our current implementation this is technically possible (although the model should learn not to do that), but we have a post-processing stage when we remove all the connected components of the molecule graph apart from the "main" one. However, it would be very easy to modify the code to explicitly mask out that choice, effectively forcing the model to choose "end of generation" if there's no position to connect to; then we wouldn't need any post-processing.
>
> **Q5**: Is it possible for a motif to be reconstructed atom by atom even if it's in the vocabulary? Are there any such cases in the training data?
>
> **A5**: It's technically possible for the model to do that, although there are no such cases in the training data by design (because if a motif is included into the vocabulary, then all of its occurrences in the training data will be generated in one-shot). We would therefore expect this behaviour to be rare; if it helps, we can compute statistics to see how often this happens in practice for various motif vocabulary sizes.

---

> > ### Comment · Reviewer_V5A5 · 2021-11-29
> > **Discussion Period**
> >
> > After reading the replies to all the reviewer's comments i maintain my score.

---

### Official Review · Reviewer_WYph · 2021-11-05

**Correctness:** 2
**Technical Novelty And Significance:** 1
**Empirical Novelty And Significance:** 3
**Recommendation:** 3
**Confidence:** 5

**Main Review:**

Strengths of the paper:
1.  The construction of the training objective for the model is reasonable.
2. The proposed method could directly start with a scaffold, which is usually used in practical drug molecule discovery.
3. The experiments show the MoLeR could generate high quality molecules comparison to a few prior methods. But there are still concerns about results. (see below)
4. The revised fragment vocabulary construction is novel and seems working well.
5. The experiment also show that generation order and vocab size is important to the quality.

Weakness of the paper:
1. The organization of the paper could be better. Specification of the model should be placed in the main paper, instead in the appendix. Now much is missing in the main method, making it difficult to understand.
2. The model using GNN+MLP is not novel. Using GNN+MLP to predict fragment, attaching atom, bond has appeared before.
3. Metrics used in the experiment is unclear (what is score and quality in Table 2 and how they are evaluated? Is the score logP or score produced by RDKit? A bit confusing).
4. The unconstrained generation and comparison to the original data is not solid to evaluate the generation capability of the model. It just evaluates the reconstruction capability. It could only serve as a validation purpose for whether the model can fit data. Figure 2 left shows the distance between generated molecules and the training set. But being able to generate similar molecules is not the goal, it would be more important to evaluate the model’s capability on generating novel molecules that are different from those in the training while satisfying desired properties.
5. Certain important baseline could be compared in the experiments, including Transformer with SMILES ([1], since LSTM is a weak baseline for generation), MARS ([2] also fragment-based approach using GNN, and can start with a scaffold and multi-property optimization).

[1] Transformer neural network for protein-specific de novo drug generation as a machine translation problem. 2018.
[2] MARS: Markov Molecular Sampling for Multi-objective Drug Discovery. 2021.

**Summary Of The Paper:**

The paper study the problem of fragment-based molecule generation.  They propose a model MoLeR which consists of an encoder of molecular graph using Graph convolutional neural network (two GNN, one for complete molecules and one for partial molecules), a MLP decoder layer to predict the fragment, the attaching atom on the fragment, and the bonding atoms on the partial molecule. To train the model, it includes three losses for multi-task learning: a KL term between variational posterior and prior of latent representation, a self-reconstruction loss, and a property prediction MSE loss.
The paper conduct experiments on GulcaMol and show that it is able to generate molecules similar to the training molecule distribution, from scratch or from a given scaffold. It shows better results than LSTM, JTVAE, CDDD-MCTS, etc.


**Summary Of The Review:**

Reasons to accept: reasonable designed training objective for the application of drug design, good generation quality, new method of constructing molecule fragment vocabulary.

Reason to reject: unclear description of method, limited novelty of model, questionable metric used in the evaluation, missing baselines.

---

> ### Author Response · Authors · 2021-11-11
> **Review response (1/2)**
>
> Thank you for your review and the constructive criticism. Find answers to each one of your concerns below (due to character limit, the answer will have two parts).
>
> **Q1**: Using GNN+MLP is not novel. Using GNN+MLP to predict fragment, attaching atom, bond has appeared before.
>
> **A1**: Agreed; using GNNs to encode molecular graphs and using MLPs to predict various choices during generation are both standard choices in the literature. In this work, we explore various ways of posing the learning problem (motif-based vs atom-by-atom, scaffold-based, lack of dependence on history, etc) and did not aim to replace the GNN/MLP architectural components (doing so just for novelty's sake did not seem useful, as it would make the insights less clear). We note that our model could easily be adapted to use another architecture to compute per-node representations, e.g., the recent Graphormer architecture, and we fully expect the insights from this paper to carry over to such a setting.
>
> **Q2**: Organization of the paper could be better. Specification of the model should be placed in the main paper instead of the appendix. Much is missing in the main paper, making it hard to understand.
>
> **A2**: Can you elaborate on which modelling details you would like moved to the main text? As you can tell from the size of the appendix, we had a lot of content and couldn't fit everything within 9 pages. We deferred the exact specification of the GNNs used to the appendix, given that (as you correctly noted) that part is rather standard and could even be seen as a hyperparameter; while we make several design choices there, these come from modern GNN literature and are not our invention, hence we felt it's not necessary to place them in the main paper (at the expense of having to remove some MoLeR-specific content).
>
> As noted above, we view these as exchangeable implementation details, with the main modelling contributions of the paper being the overall architecture that is conditioning on a single representation of a partially generated molecule, and the mix of atom and motif generation steps.
>
> **Q3**: Figure 2 just shows the distance between generated molecules and the training set, and can only serve as a validation purpose for whether the model can fit data. Being able to generate similar molecules is not the goal, it would be more important to evaluate the model’s capability on generating novel molecules that are different from those in the training while satisfying desired properties.
>
> **A3**: As noted in the paper, MoLeR yields close to 100% novel molecules (i.e., molecules not present in the training data). We thus believe the distribution matching metrics are useful as a sanity check: they show that the generated molecules cover the chemical space well (as an anecdotal example, when experimenting with CGVAE, we found the generation metrics were sub-par, which we then traced back to CGVAE underproducing aromatic rings).
>
> Overall, we agree that distribution matching metrics are not an end goal, and rather a check that the model can properly cover the chemical space. This is why most lines of experiments in our work are on molecular optimization (Table 2, Figure 4), which show that our model is able to generate valid, novel molecules with desired properties.
>
> **Q4**: Compare to: Transformer with SMILES ([R1], since LSTM is a weak baseline for generation), MARS ([R2] also fragment-based approach using GNN, and can start with a scaffold and multi-property optimization).
>
> **A4**:
> - [R1] cannot be applied to our setup, as it is a seq2seq translation model which translates from protein sequences to SMILES, while in our setup we don't assume access to protein sequences. LSTM is one of the most widely used architectures in the chemistry community (forming the basis of the REINVENT model and CDDD), and has been very successful in the lab. It therefore makes sense to have it as a baseline. Finally, while in NLP LSTMs have been largely replaced with Transformers, to the best of our knowledge there is no consensus in the generative chemistry community for whether Transformers significantly outperform LSTMs on molecular optimization.
> - MARS [R2] is an interesting paper, and we will add a citation to it. However, it is only tested on a non-standard benchmark, which has not been widely accepted in the chemistry community, and does not measure molecule quality.  As an example for a recent SotA model, which is also more directly comparable, we used MNCE-RL [1] (from NeurIPS 2020, newer than MARS), and we compared to it in Table 2.
>
> **References**
>
> [1] Reinforced Molecular Optimization with Neighborhood-Controlled Grammars
>
> [R1] Transformer neural network for protein-specific de novo drug generation as a machine translation problem
>
> [R2] MARS: Markov Molecular Sampling for Multi-objective Drug Discovery

---

> ### Author Response · Authors · 2021-11-11
> **Review response (2/2)**
>
> **Q5**: What is score and quality in Table 2 and how they are evaluated? Is the score logP or score produced by RDKit?
>
> **A5**: The first two columns (titled "Guacamol") correspond to the Guacamol benchmarks [2], as discussed in the text; the next two columns (titled "Scaffold") correspond to our own novel 4 scaffold-based benchmarks. To expand on both in details:
>
> - Guacamol [2] contains 20 multi-objective optimization tasks, which range from ones targeting rediscovery/similarity, through isomer enumeration, to scaffold/decoration hopping. This is a standard benchmark suite, used in many prior works.
> - Our new scaffold-based benchmarks were inspired by real-world clinical candidates or marketed drugs, and employ large challenging scaffolds.
>
> In both cases, the score is a task-specific real number in the [0,1] range, with higher values being better, indicating how well produced molecules match a target molecular profile. The notion of quality is the same as in Guacamol, defined as the ratio of generated molecules passing a suite of structural filters from [3]. Having high quality is extremely important, as many SotA generative models of molecules perform extremely well on score, but badly on quality. MoLeR was developed in collaboration with medicinal chemistry professionals in the pharma industry, and (anecdotally), it is the quality that makes most SotA generative models of molecules not usable in practice (it's often even desirable to slightly sacrifice score to gain on quality).
>
>
> In our scaffold benchmarks, we have 4 tasks. 3 of them are about maximizing similarity towards a target molecule (which may already have some of the required properties we care about in a drug discovery project e.g. binding), while enforcing the presence of a scaffold (which is not present in the target). This scenario reflects scaffold hopping, which is an important task in drug discovery.
>
> We use the following target molecules and scaffolds (the full code release is currently being prepared):
>
> 1)
> - target="CCCC1=NN(C2=C1N=C(NC2=O)C3=C(C=CC(=C3)S(=O)(=O)N4CCN(CC4)C)OCC)C",
> - scaffold="Cc1ncn2[nH]c(-c3ccc(F)c(S(=O)(=O)N4CCNCC4)c3)nc(=O)c12",
>
> 2)
> - target="CC1Oc2c(C)nccc2-c2cc(NC(=O)Cc3nn(C)cc13)ccc2C#N",
> - scaffold="COc1cnccc1-c1cccc(NC(=O)Cc2ccn(C)n2)c1",
>
> 3)
> - target="CC(C)C#Cc1ccc(-c2ccc(Cl)c3c(NS(C)(=O)=O)nn(CC(F)(F)F)c23)c(C(Cc2cc(F)cc(F)c2)NC(=O)Cn2nc(C(F)(F)F)c3c2C(F)(F)C2CC32)n1",
> - scaffold="CCc1cnc(-c2ccccc2)c(C(Cc2ccccc2)NC(=O)Cn2nc(CF)c3c2C(C)(F)C2CC32)n1",
>
>
> Furthermore, we have a 4th task on designing a molecule similar to Xarelto, while having the physicochemical properties of Apixaban, and also containing a required scaffold:
>
> scaffold = "CCN(C=O)C1=CC=C(C=C1)N1CCCCC1=O" apixaban = "O=C5N(c4ccc(N3C(=O)c1c(c(nn1c2ccc(OC)cc2)C(=O)N)CC3)cc4)CCCC5"
> xarelto = "O=C1COCCN1c2ccc(cc2)N3CC(OC3=O)CNC(=O)c4ccc(s4)Cl"
>
> We will also add the details presented above into the appendix of our paper.
>
> Finally, as the reviewer asks about logp optimization: some of the benchmarks in the suites above include logp as part of their multi-objectives scoring functions; in general, we believe the benchmarks investigated here to be more diverse and realistic than optimizing for logp alone (see also the Guacamol paper for a discussion).
>
> **References**
>
> [2] GuacaMol: Benchmarking Models for de Novo Molecular Design
>
> [3] https://github.com/PatWalters/rd_filters

---

### Official Review · Reviewer_2My6 · 2021-11-05

**Correctness:** 4
**Technical Novelty And Significance:** 2
**Empirical Novelty And Significance:** 3
**Recommendation:** 6
**Confidence:** 3

**Main Review:**

The authors do a great job at presenting the general challenge they target, their method, and the design choices. The paper is throughout very well-written and clear.

The presented method is very similar to work from Jin et al. (2018, 2020). The authors explicitly acknowledge this on multiple occasions and point out why the differences matter. To strengthen this point, I would welcome if the authors could elaborate a bit more on the quantitative comparison in in Figure 2. The performance of their presented method increases with the size of the vocabulary. Is this not an expected observation? There are single dashed lines for HierVAE and JT-VAE. What equivalent vocabulary sizes do these correspond to?
The authors also state that "Due to the high cost of training JT-VAE and HierVAE, we did not tune their hyperparameters and instead used the default values." This also seems to handicap these methods.

As a minor point, it would be nice if the authors could clarify whether the numbers shown in Table 1 are for the original baseline implementations or whether these already account for the code modifications that the authors made .

I was also curious about the phrase "After each choice, if the currently selected atom is part of a motif, we add the entire motif into the partial graph at once." in relation to Algorithm 2. As we move to larger vocabulary sizes, are there any individual atoms added? It would be great if these authors could elaborate on this in relation to the vocabulary sizes shown in Figure 2.

**Summary Of The Paper:**

The authors present a method for iterative small molecule generation based on an autoencoder framework with graph neural networks. The method specifically focuses on the ability to extend molecular scaffolds (predefined subparts of a molecule) with structural motifs and individual atoms. The authors show results on unconstrained molecular optimization tasks as well as tasks in which a scaffold is given.

**Summary Of The Review:**

The authors explicitly address the similarity of their work to prior contributions from Jing et al. How significant the differences are hinges in part on the quantitative comparisons. I would welcome if the authors could address my questions with respect to Figure 2 and potentially include additional comparisons.

---

> ### Author Response · Authors · 2021-11-10
> **Review response**
>
> Thank you for your review and the positive comments about our work! Below we address each one of your points one by one.
>
> **Q1**: Performance of the presented method increases with the size of the vocabulary; is this not an expected observation?
>
> **A1**: To some extent it's expected, but there are effects that could make this not the case: for example, increasing the motif vocabulary size increases the number of parameters in the model somewhat, as motif embedding parameters are motif-specific, and these additional parameters get harder to learn with more motifs (as the added motifs get more rare). We found it surprising that we could still see improvements beyond 1k motifs. To our knowledge, the impact of motif vocabulary size has not been thoroughly analysed in prior work, thus we decided to investigate it as part of our work.
>
> **Q2**: In Figure 2, what vocabulary sizes do the dashed lines for HierVAE and JT-VAE correspond to?
>
> **A2**: To address both models separately:
> - JT-VAE doesn't give an option to tweak the vocabulary size; after processing Guacamol, it ended up with 4608 motifs.
> - For HierVAE, there is some flexibility with respect to vocabulary size (although not as much as with MoLeR, because in HierVAE setting the vocab size too low would break it), but it's hard to compare sizes exactly, because HierVAE includes each motif with several configurations of attachment points (while MoLeR side-steps this complexity). We tried setting the minimum number of motif occurrences to 1000 and 100, which yielded 4317 and 7228 *structurally different* motifs respectively; this translates to 33736 and 58594 if we count all attachment point variations.
>
> In both cases, the number of motifs used is close to the largest number used by MoLeR (or far exceeds it in case of HierVAE, if we consider that different attachment point variations are treated as separate motifs).
>
> **Q3**: The authors state that "Due to the high cost of training JT-VAE and HierVAE, we did not tune their hyperparameters and instead used the default values." This seems to handicap these methods.
>
> **A3**: See "response to all reviewers".
>
> **Q4**: Are the numbers in Table 1 for the original baseline implementations or these already account for the code modifications that the authors made?
>
> **A4**: These numbers indeed include the code optimizations we made; sorry if this was not clear. Without our improvements (e.g. the multithreaded decoding we added to JT-VAE), the gap is even wider.
>
> **Q5**: I'm curious about the phrase "After each choice, if the currently selected atom is part of a motif, we add the entire motif into the partial graph at once." in relation to Algorithm 2. As we move to larger vocabulary sizes, are there any individual atoms added? It would be great if the authors could elaborate on this in relation to the vocabulary sizes shown in Figure 2.
>
> **A5**: We're not sure if we understand the question; are you asking if with large vocabulary sizes, the molecules get fully covered by motifs, and the atom-by-atom actions are not needed anymore? This is not the case: atom-by-atom mode is still required for (1) less common fragments, and (2) 1-2 atom bits that are often attached to rings (we only consider fragments of at least 3 atoms as motifs, to keep motifs non-trivial). If that helps, we'd be happy to compute some statistics, e.g. what proportion of atoms are covered by motifs for various vocabulary sizes.

---

### Author Response · Authors · 2021-11-10
**Response to all reviewers**

We would like to thank all reviewers for their efforts reviewing our work. We were particularly happy to see many positive comments made by reviewers; for example, that "the authors do a great job at presenting the general challenge they target, their method, and the design choices" and "the paper is throughout very well-written" (Reviewer 2My6); that MoLeR is "practical and pragmatic" and "strikes interesting balance of flexibility vs. first-principles" (Reviewer V5A5); and that "experimental evaluation is comprehensive" (Reviewer 9yrY). Some of the authors have experience working on real-world drug discovery, and so we arrived at MoLeR's design from the practical side, aiming to lift constraints limiting current SotA models.

We respond to each review separately, but there is one question that was repeated in two reviews, so we respond to it below to avoid duplication.

**Q1**: Hyperparameters for JT-VAE and HierVAE were not tuned, which may underestimate their performance.

**A1**: We spent considerable effort to get JT-VAE to train at all in our setting, and training using the author-provided hyperparameters took more than a month. This is one of the reasons why we are emphasizing the training performance of MoLeR, and a consequence is that we believe it is impractical to tune JT-VAE hyperparameters on new datasets. In addition, we spent several weeks trying to integrate JT-VAE with MSO as an additional baseline, but were not able to get this to work reasonably well, running into many unexpected problems (e.g., seed molecules of the optimization procedure not being encodable due to missing motifs in the vocabulary), and the overall optimization performance was below simple baselines.



HierVAE was faster to train, and while we did an initial manual exploration of some hyperparameters (varying the vocabulary size), that yielded no substantial differences in results, and so we refrained from a full exploration of the hyperparameter space.

---

### Decision · Program_Chairs · 2022-01-20

**Decision:**

Accept (Poster)

**Comment:**

Most reviewers were positive about the paper, seeing that the proposed method is practical and has convincing experimental performances. One reviewer was a bit negative and raised questions about clarity. After the authors responded, the negative reviewer didn't respond further. After reviewing all the comments, the AC feels that there is enough support from reviewers to accept this paper.